# Evaluation of Evapotranspiration Estimates in the Yellow River Basin against the Water Balance Method

**Guojie Wang \*, Jian Pan, Chengcheng Shen, Shijie Li, Jiao Lu, Dan Lou and Daniel F. T. Hagan**

Collaborative Innovation Center on Forecast and Evaluation of Meteorological Disasters, School of Geographical Sciences, Nanjing University of Information Science & Technology, Nanjing 210044, China; jianpan@nuist.edu.cn (J.P.); ccshen_nuist@163.com (C.S.); lishijie@nuist.edu.cn (S.L.); jiao_lu@nuist.edu.cn (J.L.); loudan711@163.com (D.L.); dans7messiah@nuist.edu.cn (D.F.T.H.)
\* Correspondence: gwang@nuist.edu.cn; Tel.: +86-25-5873-1418

**Abstract:** Evapotranspiration (ET), a critical process in global climate change, is very difficult to estimate at regional and basin scales. In this study, we evaluated five ET products: the Global Land Surface Evaporation with the Amsterdam Methodology (GLEAM, the EartH2Observe ensemble (E2O)), the Global Land Data Assimilation System with Noah Land Surface Model-2 (GLDAS), a global ET product at 8 km resolution from Zhang (ZHANG) and a supplemental land surface product of the Modern-ERA Retrospective analysis for Research and Applications (MERRA_land), using the water balance method in the Yellow River Basin, China, including twelve catchments, during the period of 1982–2000. The results showed that these ET products have obvious different performances, in terms of either their magnitude or temporal variations. From the viewpoint of multiple-year averages, the MERRA_land product shows a fairly similar magnitude to the $ET_w$ derived from the water balance method, while the E2O product shows significant underestimations. The GLEAM product shows the highest correlation coefficient. From the viewpoint of interannual variations, the ZHANG product performs best in terms of magnitude, while the E2O still shows significant underestimations. However, the E2O product best describes the interannual variations among the five ET products. Further study has indicated that the discrepancies between the ET products in the Yellow River Basin are mainly due to the quality of precipitation forcing data. In addition, most ET products seem to not be sensitive to the downward shortwave radiation.

**Keywords:** ET product; evaluation; water balance method; the Yellow River

## 1. Introduction

Evapotranspiration (ET) consists of water surface, soil, snow, and ice evaporation, as well as emissions from vegetation, and is a critical process in the hydrological cycle [1,2]. ET also plays an important role in the energy and carbon cycles [3], and is a significant parameter in the interactions between the atmosphere and the Earth's surface [4]. Land surface ET transfers about 60% of precipitation back into the atmosphere [5], which not only affects the land surface water balance, but also the latent heat flux [6]. It is therefore essential to accurately quantify ET for an improved understanding of the water and energy cycles. Currently, ET is mainly measured from site observations, using lysimeters [7], the Bowen ratio energy balance, and flux towers [1]. But such measurements are based on short durations and have limited spatial coverage. It remains challenging to estimate ET at the regional or basin scale, because of the sparse network of measurement sites, which have short durations and limited spatial coverage at present [8].

Over the years, several approaches have been used to estimate ET globally and regionally. Recent advances in remote sensing technology have enabled the estimation of ET within the series of ZHANG products [9], and the model tree ensemble method (MTE) has also been used to estimate ET in the Jung products [10]. Additionally, the advancement in data assimilation science has enabled the estimation of ET in different land surface model schemes, in order to produce existing products such as the GLDAS suites [11,12], and the MEERA [13] and the Japanese 55-year Reanalysis (JRA) [14] datasets as well. This has enabled us to better understand the processes in such regions at a large scale rather than point scale. For example, our understanding of the water cycle has improved with MERRA by considering various aspects of the hydrological cycle [13]. JRA is suitable for multidecadal variability and climate change, as it is the first reanalysis data with a study time covering the last half-century [14]. Among them, the MTE (model tree ensemble) product, based on eddy covariance, is derived from the in-situ observations of a machine-learning algorithm, and is considered as a high quality accurate ET product and has been widely used as a standard to validate other products. But it is limited to a relatively short period and has a sparse spatial coverage [8,15]. Remote sensing-based approaches provide reliable ET estimates at large scales by means of solar radiation, meteorological data, and vegetation indices; therefore, results may differ in different regions as a response to the different forcing data and models used [16]. Land surface models (LSMs) have been confirmed to have an excellent performance in capturing ET variation, and can also be used as an effective tool to estimate other hydrological variables [17,18]. Reanalysis data sets, derived from different forecast models or systems, can provide a comprehensive and continuous long-time series of surface hydrological variables, which are generally used to study global climate change [19]. However, because of the errors and uncertainties in the input precipitation and radiation data, the performance of the reanalysis data sets is uncertain in capturing long-term ET trends [20]. Therefore, none of the available ET products can be used as the truth, because of the uncertainties in the model structures and parameterizations.

Alternatively, the traditional terrestrial water balance method is another classical approach to estimate the ET for a closed basin, in which the ET is calculated as the residual between the observed precipitation (P), sum of the river discharge (R), and the terrestrial water storage change ($\Delta$S). The terrestrial water storage change ($\Delta$S) is very difficult to estimate, but many studies have reported that on an interannual and annual scale, $\Delta$S is very small, which is negligible when compared to the ET value. So, it is usually neglected in the water balance method for estimation at such coarse timescales of an annual period and longer [21]. During the past two decades, many studies have used this water balance method as a benchmark for data, to evaluate other data sets in different regions, such as continental china [16], the Tibetan Plateau [17,22], and around the globe [23,24]. For example, Liu conducted a global comparison of nine ET products against the water balance method during the period of 1983–2006 [8]; it was found that all of the products could not explain the long-term trends accurately, especially in the wet basins. Zhang reached a similar conclusion in 110 wet basins at a global scale [9]. Li and collaborators evaluated the seasonal changes of various ET products on the Tibetan Plateau using the water balance method [22], and found that the analyzed ET products had a great difference in performance among them. Li compared nine ET products in the middle Yellow River Basin and found that the ET derived from LSMs performed better than the reanalysis data [25].

The Yellow River basin is the sixth longest river in the world and is located in the transition zone of semi-arid and semi-humid climates. It is one of the most significant soil erosion areas in the world, especially in the middle reaches of the Yellow River, which flows across the Loess Plateau [26]. Over the past 60 years, with an increase in temperature and human activities, the occurrences of drought in the Yellow River Basin have increased significantly, which has threatened water, energy, food, and ecological security in the basin [25]. Evapotranspiration, a critical process in the hydrological cycle, is beneficial for understanding the variations in the regional water cycle, and thus, can enhance the management of water resources in the Yellow River Basin. However, the ground observation records of ET are either limited to a specific point scale in space or represent a short time span, and thus, the applications of such data may be limited in a good representation. As an alternative, several ET

products have been developed from observed, remotely sensed, and reanalysis sources to study the regional hydrological applications. Owing to the discrepancies in the input data and model structures, distinct uncertainties exist within the different products. Therefore, it is crucial to compare them in order to identify the strengths and limitations over specific regions.

The objectives of this study were as follows: (1) to evaluate five ET products (one LSM simulation, two diagnostic products, and two reanalysis-based products) against the traditional water balance method in the Yellow River Basin, and (2) to analyze the potential impacts of the forcing data on the ET products. In Section 2, we describe the study region, datasets, and methodology. Section 3 presents the evaluation results using the water balance method. Section 4 contains a discussion about the potential influences of the forcing data. In the final section, we conduct a summary and propose some future work.

## 2. Materials and Methods

### 2.1. Study Area

The Yellow River is the second largest river in China and the sixth-longest river in the world. The full length of the river basin is 5464 km and it covers an area of approximately $7.95 \times 105$ km$^2$, spanning from 95.75° E to 119.25° E and from 32° N to 42° N. Because of the complex natural conditions and undulating topography of the basin, the climate is quite different from the other drainage basins in China [27] (Figure 1). The climate of the Yellow River Basin is controlled by the East Asian monsoon, and the average annual precipitation in the whole basin is 438 mm; but, the precipitation distribution in the basin is uneven, with a decreasing trend from southeast to northwest, because of the geographical location, topography, landscape features, and vegetation cover differences across the basin [26]. The average annual precipitation in the basin is 300 mm in the northwest and about 800 mm in the southeast; precipitation in the rainy reason (from June to September) accounts for more than 60% of the annual amount, which leads to a significant seasonal runoff [28]. The annual mean temperature differs across the basin; the annual average temperature in the upper reaches of the basin is lower than 0 °C, while in the lower reaches it is above 10 °C. The Yellow River Basin has been confronted with the most serious water loss and soil erosion in the past decades [29], which shows an urgent need to improve our understanding of its water cycle. Twelve catchments are introduced to study the spatial patterns of the total ET in the Yellow River Basin. The drainage areas and coordinates of the outlets are summarized in Table 1.

**Table 1.** Locations of drainage areas of the 12 hydrological stations in the Yellow River Basin used in this study.

| No. | Station | Longitude | Latitude | Drainage Area (km$^2$) |
|-----|---------|-----------|----------|------------------------|
| 1 | Tangnaihai | 100.15 | 35.50 | 117,605 |
| 2 | Lanzhou | 103.82 | 36.07 | 222,507 |
| 3 | Jinyuan | 104.66 | 36.55 | 10,496 |
| 4 | Toudaoguai | 111.07 | 40.27 | 367,898 |
| 5 | Longmen | 110.58 | 35.67 | 497,552 |
| 6 | Hejin | 100.09 | 35.30 | 38,486 |
| 7 | Zhuangtou | 109.84 | 35.00 | 25,648 |
| 8 | Huaxian | 109.76 | 34.58 | 106,000 |
| 9 | Tongguan | 110.30 | 34.62 | 682,144 |
| 10 | Wuzhi | 113.26 | 35.07 | 12,880 |
| 11 | Huayuankou | 113.65 | 34.92 | 730,036 |
| 12 | Lijin | 118.30 | 37.52 | 751,869 |

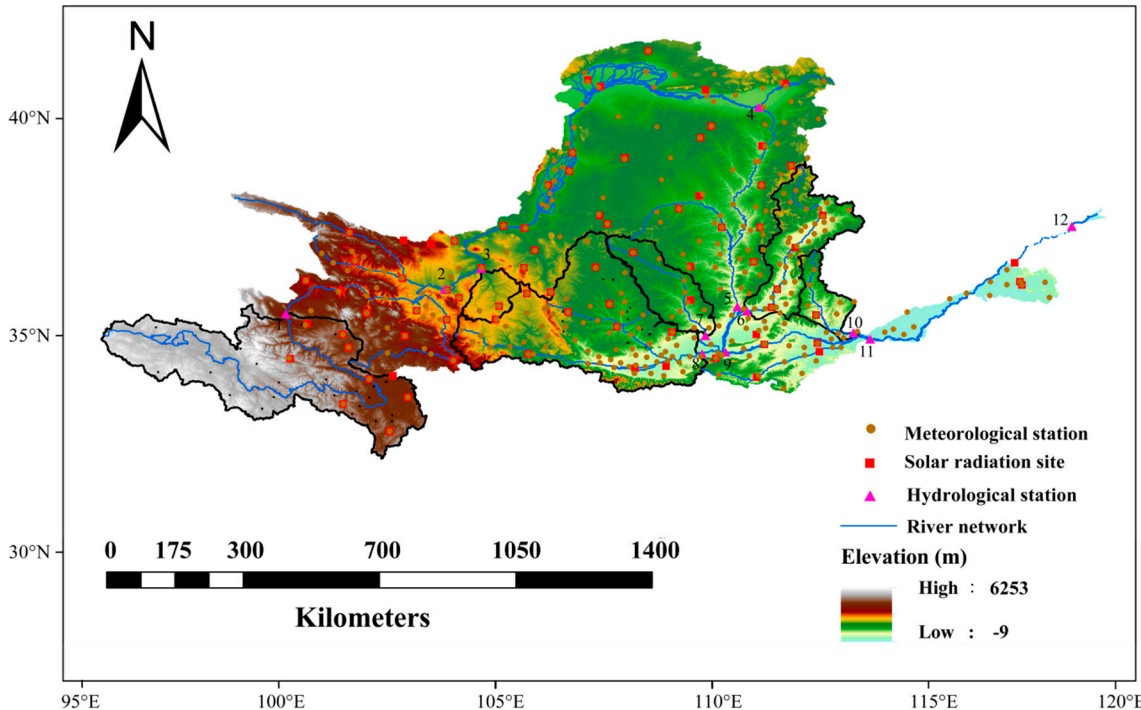

**Figure 1.** Hydrological networks and locations of the meteorological stations and solar radiation sites in the Yellow River Basin.

*2.2. Data Sets*

2.2.1. ET Products

In our study, five long-term global ET products were selected for evaluation, the details of which are presented in Table 2. They fall into the following three categories: an LSM, two diagnostic models (DMS), and two reanalysis data sets. The LSM simulation was obtained from the Global Land Data Assimilation System with Noah Land Surface Model-2 (GLDAS-Noah, version 2), with a spatial resolution of 1° [18,30].

There were two products from diagnostic models, namely: (1) GLEAM product (Global Land Surface Evaporation: The Amsterdam Methodology), which derives the different components of terrestrial ET, maximizing the use of satellite database, including transpiration, bare soil evaporation, open-water evaporation, interception loss, and sublimation, with a spatial resolution of 0.25° [31,32]. The potential evaporation was calculated using the Priestley–Taylor approach. (2) ZHANG is a global ET product at 8 km resolution for the period of 1982 to 2012. It was derived from the Numerical Terradynamic Simulation Group (NTSG, http://www.ntsg.umt.edu/project/et). It applied a modified Penman–Monteith method to estimate plant transpiration and soil evaporation, based on the Moderate Resolution Imaging Spectroradiometer (MODIS) data, meteorological observations, and satellite-based vegetation parameters [9]. This algorithm also proved to be effective to quantify open water evaporation, using the Priestley–Taylor approach.

We have also included two reanalysis-based ET products in this study. (1) E2O product: This ET data was produced by the EU-earth2observe project, aimed at developing a global water resources reanalysis database, spanning the period of 1979–2012. The E2O product consists of an ensemble of 10 ET products, which are derived from five global hydrological models, four LSMs, and a simple water balance model. The details of this ET data are referred to by Schellekens [33]. (2) MERRA_land product, which is a supplemental land surface product of the Modern-ERA Retrospective analysis for Research and Applications (MERRA). It revolves around the historical analysis of the hydrological cycle on a long-term time scale and over a wide range of climates. Compared with MERRA, MERRA_land

has the following two key improvements: first, the forcing precipitation is based on the fusion products between the NOAA (National Oceanic and Atmospheric Administration) and the MERRA precipitation, and second, the catchment model applied in MERRA_land is updated [34].

**Table 2.** Overview of the evapotranspiration (ET) products used in this study.

| Data Sets | Category | Scheme | Spatial Resolution | Reference |
|---|---|---|---|---|
| GLEAM | Diagnostic | Priestley-Taylor | $0.25° \times 0.25°$ | Miralles et al. (2011) [31] |
| E2O | Reanalysis | Fusion of 10 different ET data | $0.5° \times 0.5°$ | Jaap Schellekens et al (2017) [32] |
| GLDAS | LSM | Penman-Monteith | $1° \times 1°$ | Rodell et al. (2004) [19] |
| ZHANG | Diagnostic | Modified Penman-Monteith | 8 km | Zhang et al. (2010) [9] |
| MERRA_land | Reanalysis | GEOS-5 (Goddard Earth Observing System Model, Version 5) Catchment LSM | $0.5° \times 0.67°$ | Reichle et al. (2011) [33] |

2.2.2. Precipitation, Streamflow, Soil Moisture, and Gravity Recovery and Climate Experiment (GRACE)

In this study, we used the precipitation, streamflow, and soil moisture data from the water balance method. For precipitation, we used 19 years of observed precipitation data (Gridded Precipitation $0.5° \times 0.5°$ Grid Dataset, V 2.0), developed by the China Meteorological Administration (CMA) and the National Meteorological Information Center (NMIC, http://cdc.cma.gov.cn/). The dataset was developed from the daily precipitation records of 2474 meteorological stations across the country using thin-plate spline [35] and GTOPO30 (Global 30 arc-second elevation) DEM (Digital Elevation Model) data for reducing the influence of elevation. Different studies have used the data for climatological applications, with good degree of agreements [36]. Further details about gridded precipitation data are available in NMIC (2012). The daily streamflow data we used were measured at 12 hydrological stations during the period of 1980 to 2000, and were obtained from the Yellow River Conservancy Commission. The annual streamflow was aggregated from the daily data, which were further converted to mm/year across the respective catchment. The soil moisture data is obtained from the Land Surface Processes and Global Change Research Group (http://hydro.igsnrr.ac.cn/public/vic_outputs.html). The data is derived from the simulations of the Variable Infiltration Capacity (VIC) model, forced by meteorological observations with a three-hour time step and a $0.25°$ spatial resolution, during the period of 1952–2012, which have been demonstrated as a more reliable soil moisture data in China [37]. The Gravity Recovery and Climate Experiment (GRACE), launched on 17 March 2002, has been proven to be reliable data for estimating the terrestrial water storage change ($\Delta S$) [38,39]. The latest terrestrial water storage data (RL05) were processed at the Center for Space Research at the University of Texas (CSR), the Jet Propulsion Laboratory (JPL), and the GeoForschungsZentrum (GFZ). According to previous studies [40], we selected the data in CSR and performed stripe filtering and Gaussian filtering in order to reduce the uncertainties in estimating $\Delta S$.

The study period we selected was from 1982 to 2000, because of the temporal overlap of the used data. For the convenience of comparison, we interpolated all of the gridded data to a spatial resolution of $0.25° \times 0.25°$.

*2.3. Methods*

2.3.1. Water Balance Method

The water balance method is typically applied to estimate the ET at the basin scale [41,42], as shown in Equation (1), as follows:

$$ET_w = P - R - \Delta S \tag{1}$$

where $ET_w$ is the ET calculated by the water balance method, P is the total precipitation (mm), R is the net streamflow at the basin outlet, and $\Delta S$ is the change in terrestrial water storage (TWSC). $\Delta S$ is a very important part of the water cycle when considering its changes, but the TWSC is difficult to estimate over large areas, because of the lack of in-situ observations. Presently, the Gravity Recovery

and Climate Experiment (GRACE) is widely used to estimate TSMC in large river basins [42]. However, the GRACE data is not applicable in this study, as it starts from 2002 until the present day. Practically, ΔS is commonly assumed to be negligible at longer timescales [17,21]. However, the ΔS should be considered in our study in order to reduce the uncertainties due to the complex climate conditions in the Yellow River Basin. We used the soil moisture data derived from the VIC model simulations to estimate ΔS; studies have indicated that ΔS can be approximately estimated by the soil moisture changes within a 50 cm depth [43,44].

### 2.3.2. Error Metrics

In this study, the Bias (BIAS), Pearson correlation coefficient (CORR), and root mean square deviation (RMSD) were used as error metrics to describe the results of the assessment. They are defined as follows:

$$\text{BIAS} = \sum_{i=1}^{n} (A_i - B_i)/N \tag{2}$$

$$\text{RMSD} = \sqrt{\sum_{i=1}^{n} (A_i - B_i)^2/N} \tag{3}$$

$$\text{CORR} = \frac{\sum_{i=1}^{n} (A_i - \overline{A})(B_i - \overline{B})}{\sqrt{\sum_{i=1}^{n} (A_i - \overline{A})^2}\sqrt{\sum_{i=1}^{n} (B_i - \overline{B})^2}} \tag{4}$$

where N represents the total number of years in the study, and $A_i$ and $B_i$ represent the ET products used in this study and the water balance ET ($ET_w$) respectively.

### 2.3.3. Linear Trends

To detect the linear trends in the ET products, we used the Theil–Sen slope method [45,46]. Then, the Mann–Kendall test [47] was used to make a significance test for the derived linear trends. Both the Theil–Sen slope method and the Mann–Kendall test are non-parametric and insensitive to outliers in the time series to analyze; therefore, they have been widely used in meteorological and hydrological studies [48].

## 3. Results

### *3.1. Long-Term Averages*

Considering that ΔS is negligible in the form of long-term changes when using the water balance method, we first determined the $ET_w$ of each catchment by means of multi-year averaged precipitation and streamflow, for the period 1982–2000 (i.e., $ET_w$ = P − R), where P is precipitation and R is streamflow measured at the catchment outlet. The derived $ET_w$ indicates the annual ET of a catchment averaged throughout the study period, without considering its temporal variations.

In the following analyses, the derived $ET_w$ is used as the reference data, although it can hardly be considered as ground truth. Figure 2 shows the annual ET amount of the GLEAM, E2O, GLDAS, ZHANG, and MERRA_land products, respectively, against the derived $ET_w$ across the 12 catchments in the Yellow River Basin. It is indicated that the GLEAM and the MERRA_land products perform better than the others referred to by the $ET_w$. The MERRA_land and GLEAM show the better RMSD and BIAS, 32.51 and −20.35 mm/year for MERRA_land, and 38.42 and −33.13 mm/year for GLEAM, indicating the best agreement between the two products as well as the $ET_w$ across the 12 catchments in terms of their magnitudes. The GLDAS product shows quite similar RMSD and BIAS. In contrast, the E2O product shows large underestimations in all catchments, with the lowest RMSD and BIAS (119.95 and −133.63 mm/year, respectively), which was by far worse than the other products. The ZHANG

product is the only one that shows overestimations in most of the catchments compared with $ET_w$, with a positive BIAS of 40.67 mm/year. However, it is worthwhile to note that the results are not robustly conclusive and have great uncertainty, because of the time-span of data.

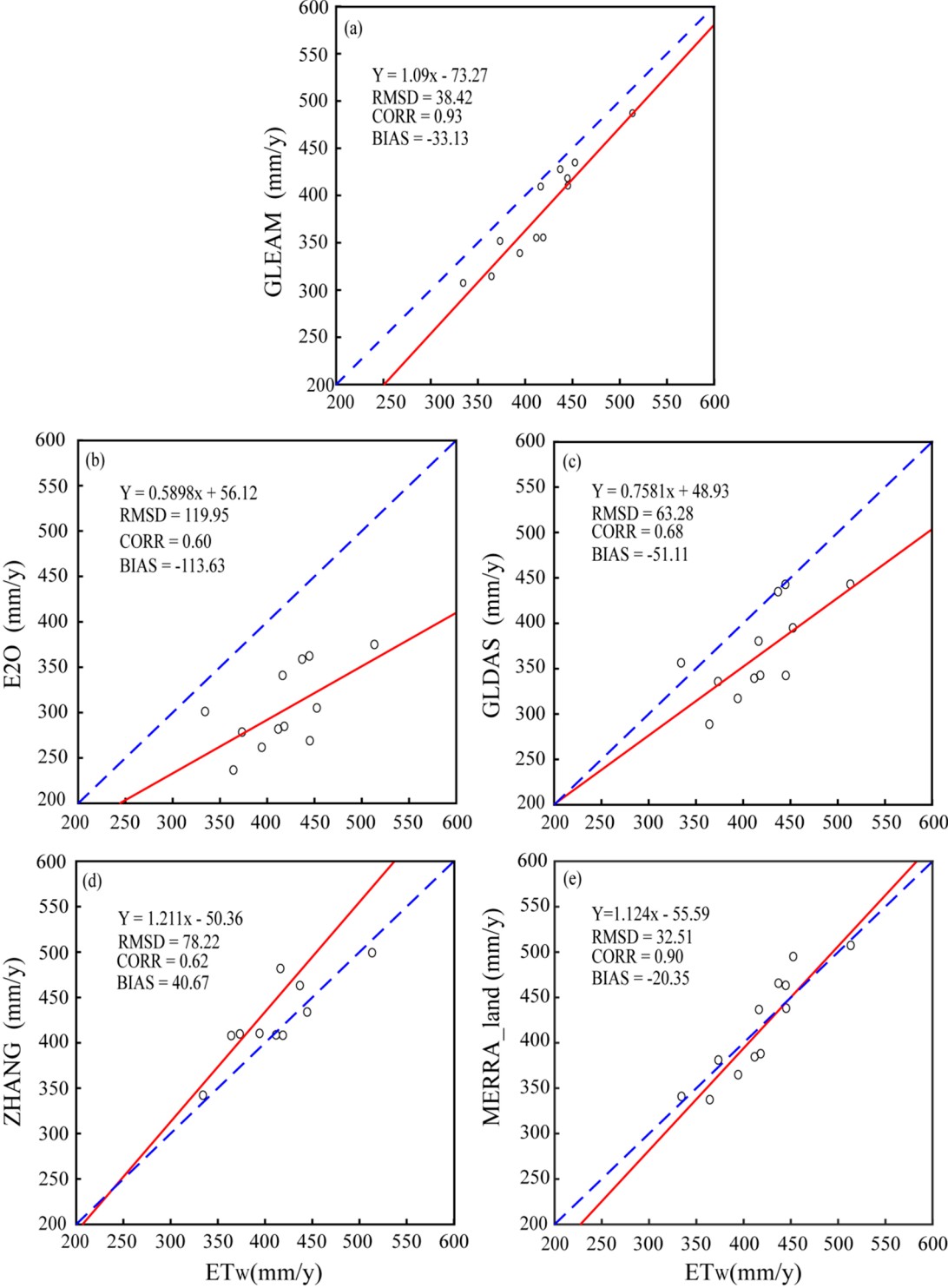

**Figure 2.** Annual evapotranspiration (ET) amount of (**a**) GLEAM, (**b**) E2O, (**c**) GLDAS, (**d**) ZHANG, and (**e**) MERRA_land against water balance ET ($ET_w$) across the 12 catchments in the Yangtze River Basin (the red line is the fitted line and the blue line is the line of perfect fit y = x).

From the perspective of spatial variations, the GLEAM product also has obvious advantages; it shows a correlation coefficient of 0.93 with $ET_w$ across the 12 catchments, indicating that the GLEAM data have a quite similar spatial pattern as the $ET_w$. However, the other three ET products have difficulty in capturing the spatial variations, particularly E2O, which shows the lowest correlation coefficient of 0.60 with the $ET_w$. It is therefore determined that the E2O product performs the worst and the GLEAM product performs the best with reference to the spatial pattern of $ET_w$.

To more clearly understand the performance of the various products at each site, we introduced the relative bias (RB) to more intuitively express the spatial distribution of each product. The RB is defined as follows:

$$RB = (ET_i - ET_W)/ET_i \qquad (5)$$

where $ET_i$ and $ET_w$ represent the ET product and that estimated from the water balance method, respectively. The results are displayed in Figure 3. Because the ET in the different regions may vary greatly, the RB can better represent the difference between the respective ET product and the $ET_w$.

In general, the results of the five ET products are significantly different. The performance of GLEAM and MERRA_land are relatively good, and show generally equivalent ET amounts to $ET_w$. Notably, in GLEAM, the largest differences occur at the largest catchments, whose outlets are the Lijin and Huangyuankou stations in the lower reaches of the Yellow River Basin, amounting to −14% and −13.89% for RB. The differences between the GLEAM ET and the $ET_w$ are smallest in the catchments in the upper reaches. This spatial pattern is consistent with precipitation, and thus the GLEAM ET and the $ET_w$ are assumed to be smaller where there is less precipitation.

The overall performance of GLDAS is similar as that of GLEAM, with underestimations compared to $ET_w$ in most catchments. However, it should be noted that the underestimations in the GLDAS ET are much larger than in GLEAM ET. Furthermore, these are by far the largest underestimations in the E2O ET, compared with $ET_w$, with the RB being larger than −30%, relatively, in most catchments. Unlike the other four products, the ZHANG product has an overestimation in most catchments. It is found that catchments with the largest differences are mostly located in the upper reaches of the Yellow River Basin.

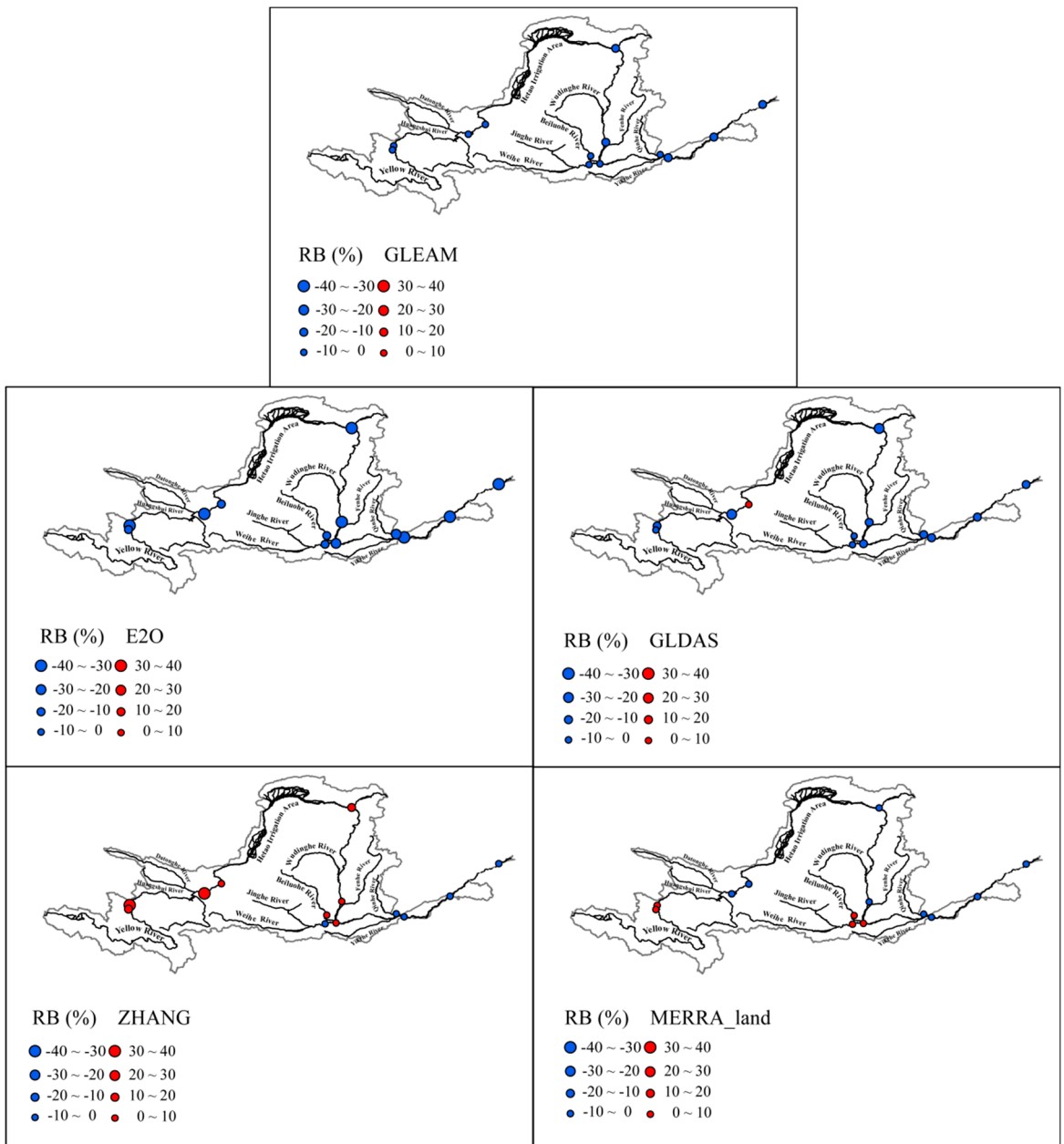

**Figure 3.** Spatial distribution of the relative bias (RB) parameter (%) between multi-year average evapotranspiration (ET) estimated from different products and from water balance ET (ET$_w$).

*3.2. Interannual Variations*

3.2.1. Estimations of Interannual ET$_w$

When calculating ET using the water balance method, it is assumed that the change in the terrestrial water storage is negligible (i.e., $\Delta S \approx 0$). Many previous studies have shown that this is feasible if the time scale is greater than one year [17,21]. However, because the Yellow River Basin experiences some of the most severe soil erosion in the world, it may be questionable to ignore $\Delta S$ in the water balance method at an interannual scale. Therefore, we have attempted to determine whether $\Delta S$ would have a significant impact on the ET$_w$ estimations. Although the GRACE data is the most commonly used to calculate $\Delta S$, its temporal coverage (from 2002 to the present day) does not have an overlap with the period of this study. We therefore used the soil moisture changes (TSMC) to approximately estimate $\Delta S$, as Lettenmaier suggested [43]. Li had proven that the soil moisture data

derived from the VIC model had a good consistency with the station data in the Yellow River Basin [25]. In order to judge the quality of ΔS simulated by the soil moisture, we also used the terrestrial water storage change (ΔS) estimated by GRACE in the period of 2004–2012.

Figure 4 shows the time series of the annual soil moisture changes (TSMC) and the differences in precipitation and runoff (P–R) during 1982–2000, as well as the comparison between the TSMC and ΔS in the period of 2004–2012. The results show that ΔS simulated by soil moisture is consistent with that estimated by GRACE as well, with a correlation coefficient of 0.86. So, this simulation is credible. The TSMC values are very small compared with the $ET_w$. The largest TSMC value is 12.19 mm, which accounts for only 2.7% of the $ET_w$, and the smallest TSMC is nearly 0. Thus, it is considered reasonable to neglect ΔS in this study when using the water balance method to calculate the $ET_w$ of each year. Therefore, the $ET_w$ is estimated with ΔS neglected in the following analysis of interannual variations.

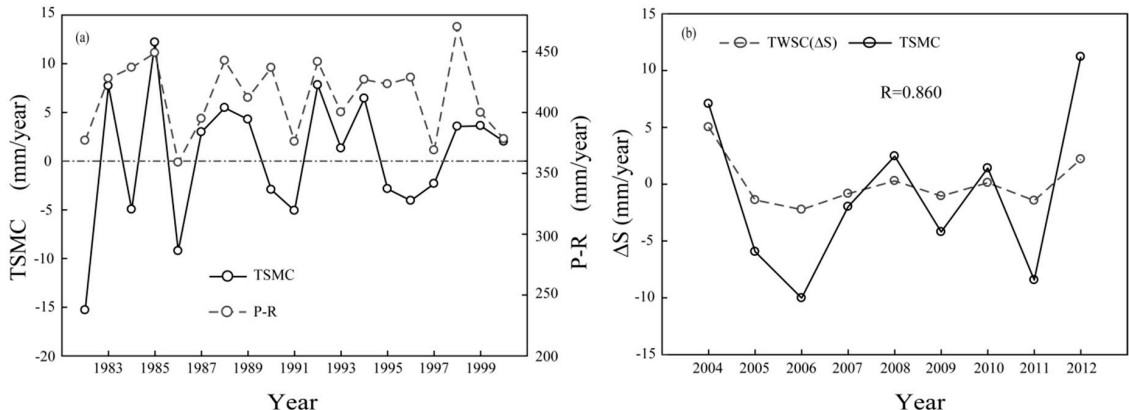

**Figure 4.** (**a**) The time series of annual soil moisture changes (TSMC) and the differences in precipitation and runoff (P–R) for the Yellow River Basin during 1982–2000, and (**b**) the comparison between the terrestrial water storage changes (ΔS) estimated by the Gravity Recovery and Climate Experiment (GRACE) and the annual soil moisture changes (TSMC) during 2004–2012.

### 3.2.2. Interannual Variations and Trends

To understand the temporal variations of land ET, we show in Figure 5 the annual time series of the five ET products and the $ET_w$ estimated from the water balance method across the Yellow River Basin controlled by the Huangyuankou station. The error metrics and linear trends of the GLEAM, E2O, GLDAS, ZHANG, and MERRA_land products against $ET_w$ across the Yellow River Basin are shown in Table 3. For 1982–2000, all of the ET products underestimated the land ET, and referred to the $ET_w$. Among them, the ZHANG product showed the best performance in terms of BIAS and RMSD (−4.91 and 29.31 mm/year, respectively), indicating that it is the best product for capturing the $ET_w$ magnitudes. In contrast, the E2O product seriously underestimated the land ET, with the largest BIAS and RMSD (−131.14 and 132.33 mm/year respectively). Similarly to the E2O product, the GLDAS product also underestimated the land ET to a large extent, but the BIAS and the RMSD were clearly superior to those of the E2O product. The GLEAM and the MERRA_land products also showed underestimations compared with the $ET_w$. However, their BIAS and RMSD were prior to those of the E2O and the GLDAS products.

However, when considering the agreement of their temporal variations, the results were quite different. The ZHANG product, which performed the best in terms of the magnitude, appeared to perform the worst in terms of the interannual variations in the period of 1982–2000, and it was the only product that did not pass the significance test at 0.05 level. In contrast, the E2O product, which showed the worst performance in terms of magnitude, appeared have the highest correlation coefficient of 0.86 with the $ET_w$, indicating that it best described the interannual variations among the five ET products. The GLEAM product also showed a good agreement with the $ET_w$ in terms of interannual

variations, with the second best correlation coefficient of 0.70. The GLDAS and the MERRA_land products show reasonable correlation coefficients (0.58 and 0.60, respectively) with the $ET_w$, indicating they can capture the interannual variations as well.

To understand their long-term changes, we estimated their linear trends using the Theil–Sen slope method and the Mann–Kendall method for a significance test; and the derived results are shown in Table 3. It shows that there was not a significant ET trend in the Yellow River Basin during the period of 1982 to 2000, except for the ZHANG product. Among the five ET products, only the trends of the ZHANG product can pass the significance test at 0.10 level, which is 10.85 mm/10a. The $ET_w$ product shows a decreasing trend of −2.35/10a, which cannot pass the significance test at 0.10 level.

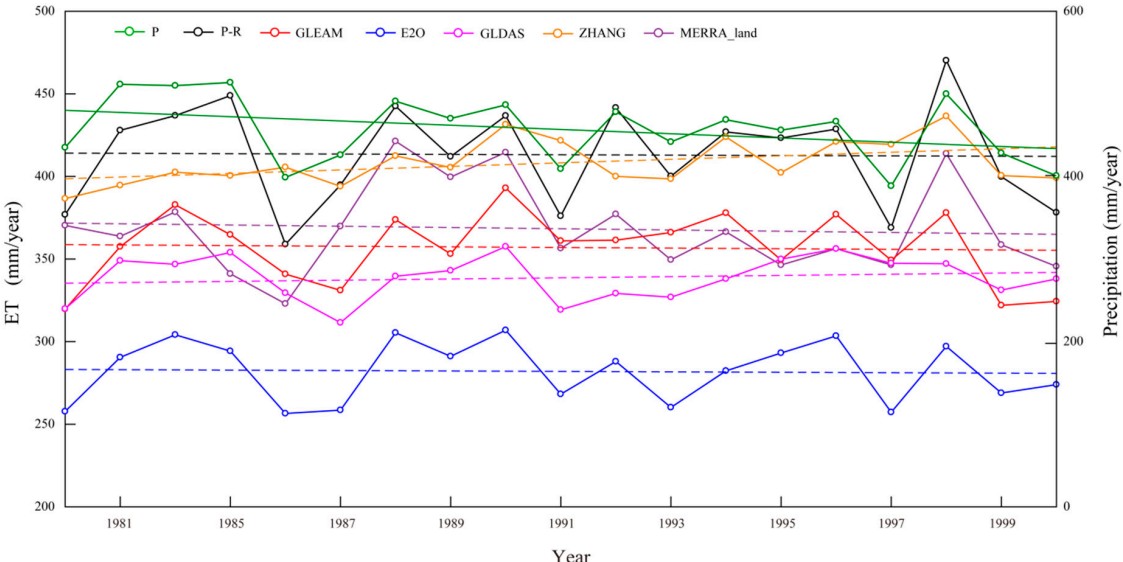

**Figure 5.** The interannual variations and trends of land ET from GLEAM, E2O, GLDAS, ZHANG, MERRA_land, and $ET_w$, respectively, across the Yangtze River Basin.

**Table 3.** Error metrics and linear trends of GLEAM, E2O, GLDAS, ZHANG, and MERRA_land products against water balance evapotranspiration (ETw) across the Yellow River Basin. The asterisk (*) indicates passing the significance test at the 0.10 level, and the asterisk (**) indicates passing the significance test at a 0.05 level.

| Product | RMSD | CORR | BIAS | MAE | Trend (mm /10a) |
|---|---|---|---|---|---|
| $ET_w$ | | | | | −2.35 |
| GLEAM | 60.22 | 0.70 ** | −56.12 | 56.12 | 0.22 |
| E2O | 132.33 | 0.86 ** | −131.14 | 131.14 | 0.67 |
| GLDAS | 78.71 | 0.56 ** | −74.49 | 74.49 | 2.83 |
| ZHANG | 29.31 | 0.33 | −4.91 | 23.23 | 10.85 * |
| MERRA_land | 51.59 | 0.60 ** | −44.77 | 44.77 | −9.08 |

## 4. Discussion

In this study, we evaluated five different ET products (two diagnostic products (GLEAM and ZHANG), one LSM simulation (GLDAS), and two reanalysis data sets (E2O and MERRA_land)) referring to the $ET_w$ derived from the water balance method in the Yellow River Basin during the period of 1982–2000. These ET products obviously have different performances, in terms of either magnitude or temporal variations. From the viewpoint of multiple-year averages, the ZHANG product shows a fairly similar magnitude to the $ET_w$ derived from water balance method, while the E2O product shows significant underestimations, accounting for only 65% of the $ET_w$.

The analysis of 12 catchments in the Yellow River Basin indicated the degree of agreement of the five ET products with the $ET_w$ in terms of their spatial pattern. The GLEAM product shows the

highest correlation coefficient of 0.93 with the $ET_w$, and the MERRA_land product shows the second highest of 0.90, indicating that both products have quite similar spatial patterns to the $ET_w$ data. The other three products show correlation coefficients ranging from 0.60–0.70, which are reasonable but much lower than the GLEAM and the MERRA_land products. From the RB spatial distribution of each product, in addition to ZHANG, the other four products are underestimated in almost every catchment. The deviation in GLEAM and MERRA_land is relatively small, and the difference in GLEAM may be related to precipitation. The more precipitation, the smaller the difference. The overall performance of GLDAS is similar as that of GLEAM, and the best performances are found in the catchments in the middle reaches of the Yellow River Basin. ZHANG is overestimated in most of the catchments and is significantly overestimated in the upper reaches of the Yellow River, probably due to topographical factors [9], as the upper reaches of the Yellow River are the lowest elevations. Moreover, in general, independent small catchments' performance is better than larger catchments, especially in E2O. This phenomenon may be attributed to there being more reservoir and dam construction and water diversion for irrigation in the mainstream, which may lead to more errors in the runoff data [49,50].

When studying the interannual variations, we used the soil moisture change derived from the VIC model to approximately estimate the water storage change; and, it is found to account for only about 3% of the $ET_w$, which is calculated from precipitation minus streamflow. This justifies that the water storage change ($\Delta S$) can be neglected when using the water balance method to estimate ET at interannual time scales in the Yellow River Basin. It is interesting to find that, although the ZHANG product shows the best agreement with the $ET_w$ in terms of magnitude, it appears to have a poorest agreement with the $ET_w$ in terms of interannual variations. In contrast, although the E2O product is by far the largest deviation in magnitude, it best agrees with $ET_w$ regarding the interannual variations with a highest correlation coefficient of 0.86. Generally speaking, the GLEAM product shows a fairly similar magnitude, spatial pattern, and also temporal variations as the $ET_w$ data; therefore, it can be considered a rational ET product in the Yellow River Basin.

*Possible Reasons for the ET Product Performance*

There are various reasons for the discrepancies in the performance among the ET products, including the model or algorithm types and the forcing meteorological data, as well as the surface cover data [51–54]. Many previous studies have proven that the forcing data plays a decisive role in the performance of ET products [8,25,55,56]. In this study, we selected the two most important forcing variables (i.e., precipitation and downward shortwave radiation) for further analysis, so that their potential impacts on product uncertainty could be understood.

For the GLEAM product, the precipitation is the Multi-Source Weighted-Ensemble Precipitation (MSWEP: $0.25° \times 0.25°$, 1980–2000), and the downward shortwave radiation is extracted from the ERA-interim reanalysis [32]. The GLDAS product uses the latest NOAH2 model that was forced entirely with the Princeton meteorological forcing data [18]. The MERRA_land reanalysis data benefits from corrections to the precipitation forcing data using the global gauge-based NOAA Climate Prediction Center Unified (CPCU) precipitation product [34]. As for the E2O product, the precipitation and radiation from the Integrated Project Water and Global Change (WATCH) forcing data are used [33]. The radiation data of the ZHANG product is obtained from the NASA World Climate Research Program/Global Energy and Water-Cycle Experiment (WCRP/GEWEX) Surface Radiation Budget (SRB) Release 3.0 dataset [30].

The annual precipitation from the China Meteorological Administration (P_CMA) and the downward shortwave radiation from the Institute of Tibetan Plateau Research, Chinese Academy of Sciences (Ra_ITPCAS, http://dam.itpcas.ac.cn/chs/), which integrates radiation from 719 ground stations (79 solar radiation sites in the Yellow River Basin) and has been validated to be credible throughout China [17], are used as baseline data to assess the feasibility of various forcing datasets.

The annual precipitation and shortwave downward radiation across the Yellow River Basin used in five ET products are shown in Figure 6. Their error metrics, referred to as P_CMA and Ra_ITPCAS, are summarized in Table 4. We can see that, the underestimations of GLEAM, GLDAS, and E2O ET have largely resulted from the negatively biased precipitation, especially for E2O. The better performance of GLEAM and E2O in capturing the interannual variability of $ET_w$ may be strongly related to their higher quality precipitation forcing data. Similarly, the worst performance for ZHANG in interannual variability may mainly result from neglecting precipitation. For GLDAS, although it has a relatively good precipitation forcing data, it still does not perform well in interannual variability, indicating that the precipitation may not play a decisive role in the GLDAS model. The precipitation used in MERRA_land is not in good agreement with the CMA precipitation in the Yellow River, which have been reported by many previous studies [8,57].

From the perspective of radiation, all products, except for E2O, have a good correlation coefficient (>0.7) with Ra_ITPCAS, and that for ZHANG is the best. However, the ZHANG product has the poorest performance in capturing the interannual variability, indicating that the shortwave downward radiation does not seem to be a major controlling factor in the Yellow River Basin. Furthermore, the radiation is overestimated to a large extent in the GLEAM and the E2O forcing data, compared to the Ra_ITPCAS; however, the annual mean ET shows the opposite pattern, indicating that the GLEAM and the E2O products are not sensitive to radiation. The radiation quality of GLDAS is relatively good, but it is difficult to judge whether it has a significant impact on the ET product. More specially, the radiation forcing data for MERRA_land shows an obvious underestimation, which may have led to the underestimated ET in MERRA_land product [6,58].

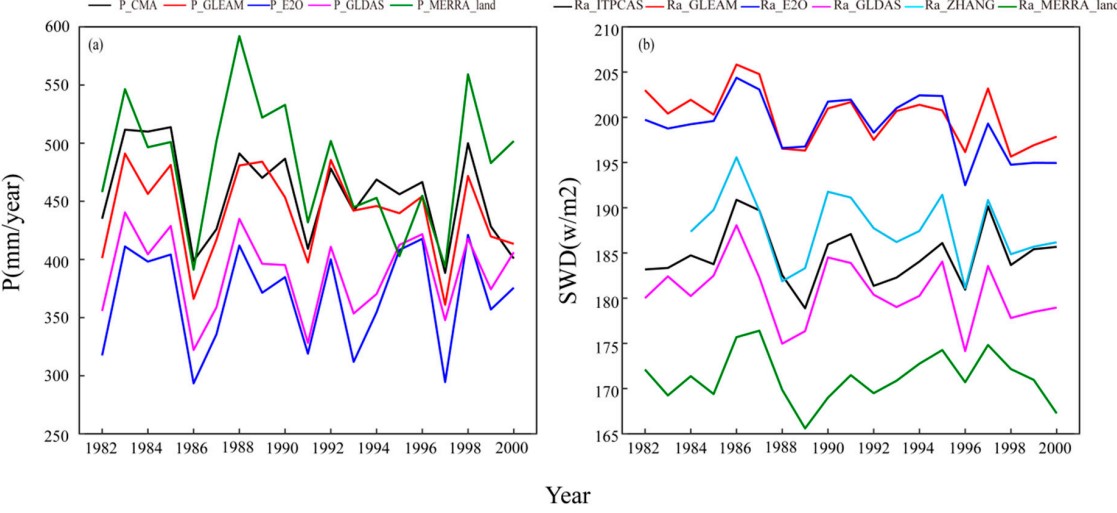

**Figure 6.** (**a**) The annual precipitation and (**b**) the shortwave downward radiation across the Yellow River Basin used in the GLEAM, E2O, GLDAS, and MERRA_land products.

**Table 4.** Error metrics of precipitation and radiation data across the Yellow River Basin in the GLEAM, the E2O, the GLDAS, the ZHANG, and the MERRA_land products, referred to P_CMA and Ra_ITPCAS, respectively.

| Data Sets | Precipitation | | | Radiation | | |
|---|---|---|---|---|---|---|
| | RMSD | CORR | BIAS | RMSD | CORR | BIAS |
| GLEAM | 24.02 | 0.90 | −16.80 | 15.53 | 0.74 | 15.38 |
| E2O | 92.80 | 0.80 | −89.11 | 14.65 | 0.56 | 14.35 |
| GLDAS | 72.87 | 0.78 | −68.48 | 4.90 | 0.75 | −4.10 |
| ZHANG | - | - | - | 3.70 | 0.78 | 2.88 |
| MERRA | 48.08 | 0.67 | 25.66 | 13.65 | 0.74 | v13.48 |

To better distinguish the effects of the two factors of precipitation and radiation, we also analyzed the monthly time series of runoff and precipitation (in Figure 7) and the relationship between ET and its corresponding precipitation and radiation forcing data, as listed in Table 5. The result shows that the runoff data is consistent with the precipitation and has obvious seasonal characteristics, indicating that the precipitation has a significant role in the water cycle in the Yellow River Basin. In Table 5, all of the ET products has a significantly high correlation coefficient (>0.60) with the precipitation forcing data, except for ZHANG (ZHANG does not use precipitation forcing data, and the listed correlation coefficient is calculated from ZHANG ET and P_CMA). However, for all of the ET products, their correlation coefficients with radiation are rather small and not significant. So, we have reason to suspect that precipitation has a more important role than radiation in the Yellow River Basin, which is consist with previous studies [25,55]. Many studies have also reported that the performance of all of the ET products, including the magnitudes, interannual variability, and linear trends, is mainly dependent on the quality of precipitation forcing data in the Yellow River Basin [8,17]. It is worth noting that although solar shortwave downward radiation does not seem to be a significant factor in the Yellow River Basin, it has been recognized as a key predictor in wet regions, where the impacts of water supply are instead not significant [59,60]. Studies have indicated that the quality of radiation observations from satellites may hinder the attributions of the ET changes to radiation [54], and many studies have found that the insufficient accuracy of the remotely sensed radiation may have indeed caused uncertainty in the ZHANG ET [8,23,25].

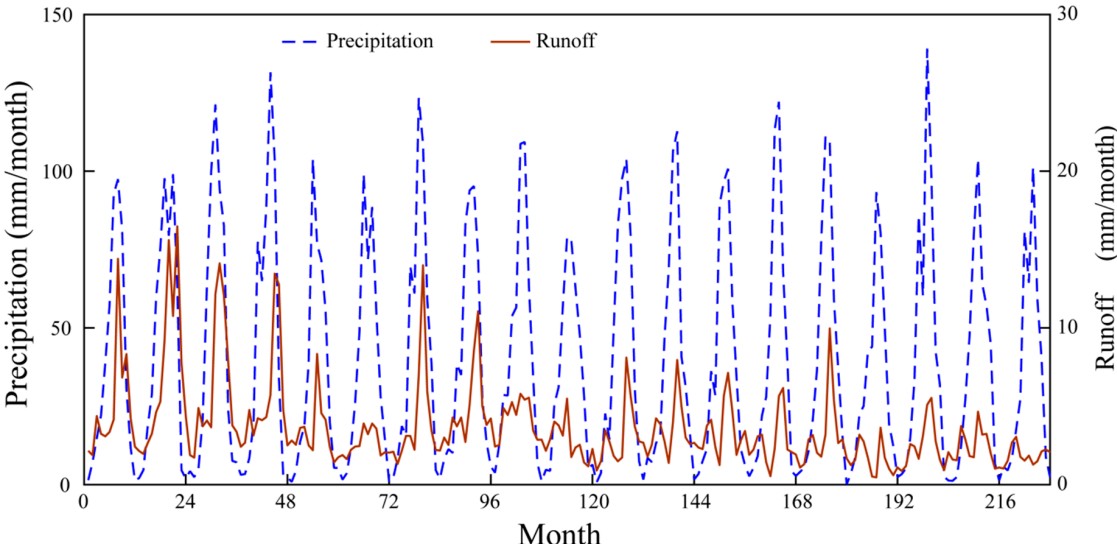

**Figure 7.** The monthly precipitation and runoff data across the Yellow River Basin.

**Table 5.** Pearson correlation between the ET products and the corresponding precipitation and radiation forcing data. The ** and * indicate significant levels at 0.05 and 0.01 levels, respectively.

| Variable | GLEAM | E2O | GLDAS | ZHANG | MERRA |
|---|---|---|---|---|---|
| Precipitation | 0.61 * | 0.89 ** | 0.67 ** | 0.12 | 0.90 ** |
| Radiation | −0.23 | −0.39 | −0.13 | −0.06 | −0.37 |

## 5. Conclusions

The significance of the Yellow River Basin to both the settlements over the region and China as a whole, as well as to the climate, has been shown in previous studies and in this study. ET, as a climate process, has a distinct impact on the hydrologic and energy cycle of the basin, consequently impacting the available water over the region. The difficulty in measuring ET and the lack of available sufficient observations has made the use of additional sources of ET information from reanalysis and satellite

products a very important option. However, different sensitivities of these additional existing products to climate variables, such as precipitation and net radiation, has created the necessity to evaluate them and understand their strengths and differences. In this study, three commonly used independent ET datasets over the basin, the ZHANG, GLDAS, and MERRA_Land, are evaluated against the water balance method to understand how they capture ET processes. Additionally, the GLEAM and E2O, which have never been evaluated in the region, were also used in this study. While all of the datasets were found to have captured the temporal dynamics of the ET over the basin, the GLEAM product showed significantly close temporal variabilities to the ET estimated with the water balance method. The MERRA_land product performed the best in describing the spatial characteristics, and the GLDAS product had a satisfactory performance as well. The ZHANG product was also found to show very close magnitudes to the reference ET, while the E2O product best described the interannual variations. Because of the limited data available, uncertainties in the representation of the long-term climatology of ET in the basin still remain. With the continual production of additional data, future studies will be able to provide more credible information to this field.

**Author Contributions:** Conceptualization, G.W. and J.P.; Methodology, J.P. and S.L.; Data Curation, S.L. and J.L.; Visualization, C.S.; Supervision, D.L. and D.F.T.H.; Writing-Review & Editing, G.W. and J.P.

**Funding:** This research was funded by the National Key Research and Development Program of China (2017YFA0603701), the National Natural Science Foundation of China (41605042, 41561124014, and 41375099), the Postgraduate Research and Practice Innovation Program of Jiangsu Province (KYCX17 0888), and the Jiangsu Meteorological Bureau 2014 modernization project "Jiangsu ocean meteorological integrated service system".

**Acknowledgments:** This study is supported by the National Key Research and Development Program of China (2017YFA0603701), the National Natural Science Foundation of China (41605042, 41561124014, and 41375099), the Postgraduate Research and Practice Innovation Program of Jiangsu Province (KYCX17 0888), and the Jiangsu Meteorological Bureau 2014 modernization project "Jiangsu ocean meteorological integrated service system". We thank International Science Editing (http://www.internationalscienceediting.com) for editing this manuscript.

**Conflicts of Interest:** The authors declare no conflict of interest.

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
