# Peer review of "Evaluation of Evapotranspiration Estimates in the Yellow River Basin against the Water Balance Method"

_water, doi:10.3390/w10121884_

Round 1
Reviewer 1 Report
Dear Authors,
The proposed research in evaluating the estimates of Evapotranspiration in the Yellow River Basin seems to be very relevant while the world is picking up drastic climate change. However, there are potential facts are missing in this paper as mentioned in the file enclosed.
Respecting the work of authors, I strongly recommend you to revise the manuscript as indicated. Hoping to see your revised one, I appreciate your effort.
I look forward to see the revised version of the paper.
Sincerely
Anonymous Reviewer.

Author Response
We are grateful to all the reviewers for their helpful comments to improve the manuscript. All the comments have been carefully read, and we have responded to them as suggested by the reviewers.
Comments from reviewer 1:
The proposed research in evaluating the estimates of Evapotranspiration in the Yellow River Basin seems to be very relevant while the world is picking up drastic climate change. The abstract is well written and attractive for the general audience. The effort made in this research is very interesting and also the manuscript has fewer mistakes on the usage of English semantics.
Responses: We appreciate this comment.
1) First attention of the reviewer goes in to the structure of the structure of the paper aligned with the template as suggested by the journal but it lacks conclusion section which is critical part of the scientific paper.
Responses: We acknowledge this comment. We have added a conclusion section in the revised manuscript to make it more aligned with the template. Please refer to the conclusion section line 442-453.
2) The major weakness of the manuscript is the lack of concrete objectives and the motivation of the research. The authors mentioned in paragraph staring from line 66, the importance of the study just because of the features of the Yellow River Basins. At this moment, it does not mentioned properly why the other studies are not working. Want are the limitations of other approach. What is the novelty of the approach? For the reviewer, just the comparison of different product is not the novel thing to try.
Responses: We are grateful for these valuable insights and comments of the reviewer. In the revised manuscript we have concisely added the motivation, novelty and compared with the limitations of the previous studies. Details of the revisions are listed below as well as added to the revised manuscript in the relevant sections respectively.
a) Firstly, we have introduced our motivation in line 81-89. It now reads, “Evapotranspiration, a critical process in the hydrological cycle, is beneficial to understanding variations in the regional water cycle and thus, enhance the management of water resources in the Yellow River basin. However, the ground observation records of ET is either limited to a specific point scale in space or represent a short time span and thus, the applications of such data may be limited in a good representation. As an alternative, several ET products have been developed from observed, remotely sensed and model reanalysis sources to study the regional hydrological applications. Owing to the discrepancies in the input data and model structures, distinct uncertainties exist within the different products. Therefore, it is crucial to compare them to identify the strengths and limitations over specific regions. ”
b) why the other studies are not working?
Unfortunately, we are not entirely sure of the intentions of the reviewer from this comment but we interpret it as, “why we think previous studies are not enough to cover the goal of this study”. With the development of new products with more sophisticated algorithms and model structures, it is necessary to compare the performance of these recent products with existing studies. Given that the GLEAM products are continuously updated, and the recent development of the E2O products in the recent years, this becomes the first study to include them in an evaluation study over Yellow River. Thus, the current study will be serving as an extension to add more valuable findings to the existing domain of previously listed studies.
c) What are the limitations of other approach.
There are many approaches to measure ET focused on point observations, such as eddy covariance systems, Bowen ratio energy balance systems and lysimeters. However, such measurements are based on short durations and have limited spatial coverage (Mao et al.,2017; Li et al.,2014). In the current study, the ET is estimated with traditional water balance method, which has been reported to be a relatively good method for estimating ET at regional or basin scales.
d) What is the novelty of the approach?
We used observed runoff, and precipitation data for estimating the ET at a relatively larger scale using the water balance method for multiple sub-basins in the Yellow River. Secondly, we not only compare the ET products, but also study the impact of the main forcing data (precipitation and radiation) in each product. As mentioned earlier, two new products that have not been evaluated over this region have been included.
3) Another point is that the methodology section does not cover all the approaches that are used in this study. For example, coming to the line 355, the authors mentioned that downward shortwave radiation as one of the forcing factor that the authors used in this study. But, the whole manuscript is about precipitation, runoff and ET. How, at this point, authors came to include the downward shortwave radiation since all the analysis was focused on precipitation only.
Responses: We appreciate the reviewer’s comments. The main purpose of this study is about the performance of various ET products in the Yellow River Basin. Since precipitation and the downward shortwave radiation are forcing datasets influencing the performance of each product, we have included the role of downward shortwave radiation in analyzing its effect ET estimation. This can be seen in the revised manuscript on lines 401-410 as,
“From the perspective of radiation, all products, except for E2O, have a good correlation coefficient (>0.7) with Ra_ITPCAS and that for ZHANG is the best. However, the ZHANG product has the poorest performance in capturing the interannual variability, indicating that the shortwave downward radiation does not seem to be a major controlling factor in the Yellow River Basin. Besides, the radiation is overestimated to a large extent in the GLEAM and the E2O forcing data, compared to the Ra_ITPCAS; however, the annual mean ET shows the opposite pattern, indicating that the GLEAM and the E2O products are not sensitive to radiation. The radiation quality of GLDAS is relatively good, but it is difficult to judge whether it has a significant impact on the ET product. More specially, the radiation forcing data for MERRA_land shows an obvious underestimation, which may have led to the underestimated ET in MERRA_land product [57]. ”
We also introduced the method of calculating radiation in line 386-389. The reason it is not shown in section 2.2 is because the water balance method does not requre this input for estimating ET. The revised section at Line 386-389 can be read as,
“The annual precipitation from the China Meteorological Administration (P_CMA) and the downward shortwave radiation from the Institute of Tibetan Plateau Research, Chinese Academy of Sciences (Ra_ITPCAS) (http://dam.itpcas.ac.cn/chs/), which integrates radiation from 719 ground stations (79 solar radiation sites in the Yellow River Basin)”
4) This paper has nice part of discussion. But it lacks concrete conclusion and recommendation of future avenues. The reviewer and readers in future want to explore and know about more about this research. As mentioned earlier, lack of conclusion and future directions suggested hints us the limitation of the approach. More specifically, there is no new contribution to the community since it just compares the result from different product. Even though mass balance model presented is here is also relies on simplified assumptions.
Responses: In response to the comment from reviewer, we have added conclusions and proposed future work directions in the revised manuscript in line 442-453, which can be read as,
“Although there are some uncertainties in this study, such as the short temporal domain, the quality of the precipitation data and the assumption of neglecting ΔS, it is still very essential and meaningful for us to better understand the hydrological cycle and climate change and it is beneficial to more effectively manage and schedule water resources in the Yellow River. Furthermore, the use of independent sources of ET datasets allows us to quantify the additional contributions from the passive microwave sources such as in the GLEAM, the MODIS source in the ZHANG, and the different independent reanalysis schemes of GLDAS, E2O and MERRA_Land. The results show that, in general, there are many discrepancies in the performance of five ET products with several error metrics, but certain product like GLEAM performed relatively better than other products. And in the future work, we should pay more attention on the uncertainties in the ET models and improving the quality of forcing datasets, as the better forcing dataset is necessary to produce more credible ET products with better performance.”
The future direction can be read as “it is still very essential and meaningful for us to better understand the hydrological cycle and climate change and it is beneficial to more effectively manage and schedule water resources in the Yellow River. Furthermore, the use of independent sources of ET datasets allows us to quantify the additional contributions from the passive microwave sources such as in the GLEAM, the MODIS source in the ZHANG, and the different independent reanalysis schemes of GLDAS, E2O and MERRA_Land.” (L443-448).
5) In line 27-28, authors mentioned that ET is a crucial parameters in the interactions between the atmosphere and the earth surface. However, to the reviewer, subsurface also plays an equally important role, so it needs to restate. What about sub surface?
Responses: We appreciate the reviewer’s time and inputs, which indeed will improve the quality of this work. Of course, subsurface also plays an important role in the hydrological cycle. But some ET products we used in the work such as GLEAM and ZHANG, They calculate the evapotranspiration of the Earth's surface, so for a fair comparison, we mainly emphasize the land surface here. And many articles have similar description, like “ET has been defined as water loss from the Earth’s surface to the atmosphere, and it is associated with soil and surface water evaporation and plant transpiration” (Mao et al., 2017) and “Evapotranspiration (ET) is a critical process that determines terrestrial water budget and exchanges of surface energy”(Li et al., 2014) and “Evapotranspiration (ET) is the flux of water transferred from the land surface to the atmosphere” (Zeng et al., 2014)
6) What the various ET products are as mentioned in line 39. Please provide appropriate examples.
Responses: In response to the comment of the reviewer, we have included details of the ET products in the revised manuscript in line 38-43 which can be read as,
“Over the years, several approaches have been used to estimate ET globally and regionally. Recent advances in remote sensing technology has enabled the estimation of ET within the series of ZHANG products [9], and the model tree ensemble method (MTE) has also been used to estimate ET in the Jung_E products [10]. Additionally, the advancement in data assimilation science has enabled the estimation of ET in different land surface model schemes to produce existing products as the GLDAS suites [11,12], MEERA [13] and JRA [14] datasets as well.”
7) The statement made in line 39, "our understanding of the global water cycle has become more profound”is a kind of doubtful. If so, whey recent advancement is going. Please be careful while making such statement. Moreover, this statement is contradicted with the statements made in the end of this paragraph.
Responses: We acknowledge the reviewer inputs and time for reviewing the manuscript which indeed will improve the quality of the work. In response to the comment about the word profound, we have elaborated the context for using this word in the lines 38-45, which can be read as,
“over the years, several approaches have been used to estimate ET globally and regionally. Recent advances in remote sensing technology has enabled the estimation of ET within the series of ZHANG products [9], and the model tree ensemble method (MTE) has also been used to estimate ET in the Jung_E products [10]. Additionally, the advancement in data assimilation science has enabled the estimation of ET in different land surface model schemes to produce existing products as the GLDAS suites [11,12], MEERA [13] and JRA [14] datasets as well. This has enabled us to better understand processes in such regions at large scale rather than point scale with more realistic and physical based algorithms and methods approaches.”
8) Why the terrestrial water storage change is assumed to be negligible at coarser time scale as mentioned in lines 56-57? The authors need to clarify the facts prior to stating like this.
Responses: We acknowledge the reviewers comment for improving the quality of the work. In the modified manuscript we have revised the section in line 63-66 which read as,
“The terrestrial water storage change (ΔS) is very difficult to estimate, but many studies have reported that on interannual and annual scale ΔS is very small, which is negligible when compared to the ET value. So it is usually neglected in the water balance method for estimation at such coarse timescales of annual and longer.”
9) What are those observations difficult to obtained in the field as mentioned in line 158?
Responses: Thank you very much for this comment. There are many difficulties for those observations which measure ΔS. Firstly, The selection and laying of the site is time consuming, labor intensive and, due to technical and economic reasons, the amount of data acquired is limited. So the number of stations is limited. Secondly, due to uneven distribution of sites, a proper spatial coverage is difficult to retrieve.
10) Figure 2 is clean and easy to visualize. But it needs definition of colors and point styles explicitly. It also needs to indicate (a) (b) and so on in the graph as well for effective presentation.
Responses: We acknowledge that indeed the Figure. 2 lake these points as raised in the comment. In response to the comment we have made thorough changes to Figure 2, as can be seen in the revised manuscript, line 229
Figure 2 Annual ET amount of a) GLEAM, b) E2O, c) GLDAS, d) ZHANG, e) MERRA_land against ETw across the 12 catchments in the Yangtze River Basin (the red line is the fitted line and the blue line is the line of perfect fit y=x).
11) What are the two indicators in line 230? Please list them explicitly.
Responses: We acknowledge the reviewer’s inputs for indicating this point which indeed is confusing and misleading. In the original manuscript, the two indicators represented BIAS and relative bias (RB). In the revised manuscript, we have removed the result of BIAS (in the original manuscript Table 3), because the results of two indicators are too similar. The results of RB are shown in Figure 3 instead, which is given in line 246-259 and can be seen as follows,
Figure 3 Spatial distribution of the RB parameter (%) between multi-year average evapotranspiration (ET) estimated from different products and from water balance ET (ETw).
12) In order to say that VIC model was better in study in [25] so used in this study, it requires popper justification if the studies are done for same purpose. Explain implicitly.
Responses: We thank this reviewer very much for these suggestion, which can greatly improve this study to be more complete and rigorous. According to the suggestion, we have added some justification in the revised manuscript before we use it. It is in the line 274-276, which reads “And Li [25] had proven that the soil moisture data derived from the VIC model had a good consistent with the stations data in the Yellow River Basin.”
13) While the E2O product show the worst performance in terms of the magnitude even with the highest correlation coefficient, but why it best describes the interannual variation? This statement is a kind of contradictory.
Responses: Thank you for your comments. Before we conducted this study, we have reviewed numerous papers, in which the“interannual variation”was used to determine the consistency of the two product in time variation, such as “interannual variation of ET from Zhang_E was poor, as indicated by a negative correlation coefficient”(Xue et al., 2013) and“In contrast, interannual variation of AETZhangKe was poorly associated with the lowest CORR”(Li et al., 2017). So magnitude and interannual variation are two separate parts, and there is no contradiction between them.
14) It is good physical reasoning for the performance of GLEAM, E2O, ZHANG, and MERRA_land. One thing is that the reasoning mentioned here are not completely convincible. For this, the physical explanation of each model is not mentioned. Second, this is what about GLDAS. Please be consistent and clear in the paragraph starting from line 324.
Responses: We thank this reviewer very much for these suggestion, which can greatly improve this study to be more complete and rigorous. We have already included an overview of a physical reasoning for the observed performances of the products. Please refer to lines 344-353.
“The deviation in GLEAM and MERRA_land is relatively small, and the difference in GLEAM may be related to precipitation. The more precipitation, the smaller the difference. The overall performance of GLDAS is similar as that of GLEAM and the best performances are found in the catchments in the middle reaches of the Yellow River Basin. ZHANG is overestimated in most of the catchments and is significantly overestimated in the upper reaches of the Yellow River, probably due to topographical factors [9], as the upper reaches of the Yellow River are the lowest elevations. Moreover, in general, independent small catchments performance is better than larger catchments, especially in E2O. And this phenomenon may be attributed to there being more reservoir and dam construction and water diversion for irrigation in the mainstream, which may lead to more errors in runoff data [48,49].”
15) The annual precipitation and downward shortwave radiation shown in Figure 6 shall either be catchment specific or need to mention how such quantities are calculated fie entire basin.
Responses: Thank you for your comments. The original annual precipitation and downward shortwave radiation data were spatially interpolated from 2472 meteorological stations and 719 solar radiation sites in the whole China. Among them, there are 307 meteorological stations and 79 solar radiation sites in the investigation area. And then, we intercepted the Yellow River Basin data for comparison.
16) MTE introduced in line 40 is not clearly understood as explained in subsequent lines.
Responses: Thank you for your comments. We also found that the original introduction to MTE is not specific enough, and we have already expanded this part in line 45-48, which reads“The MTE (Model Tree Ensemble) product, based on eddy covariance, is derived from in-situ observations by a match-learning algorithm and it is considered as high quality ET product and have been widely used as a standard to validate other products. But it is limited to a relatively short period and has a sparse spatial coverage”. And we also introduce some articles to help us explain MTE product (Wang et al.2012; Liu et al.2016).
17) The sentence made in line 59 is not semantically correct. Sentence with citation number cannot treat the number as subject. Rather it is suggested to use "X or X and collaborators [#] ...." Similar issues are seen at several places. Authors are requested to avoid such mistakes and rewrite accordingly.
Responses: Thank you very much for this suggestion. We are sorry that this is an obvious format issue and we have corrected all such issues. For example, in the revised manuscript line 68-73, it reads “Liu [8] conducted a global comparison of nine ET products against the water balance method during the period of 1983-2006; it is found that all products could not explain the long-term trends accurately, especially in the wet basins. Zhang [9] reached a similar conclusion in 110 wet basins over the globe. Li and collaborators [22] evaluated the seasonal changes of various ET products on the Tibetan Plateau, using the water balance method”
18) Having table first in Section 2.2.1 seems to be awkward, therefore such table shall be embedded somewhere inside the text below.
Responses: Thanks. This is a very good suggestion and we have adjusted the position of Table 2 to the back of the text in line 151.
19) Put subscript for A and B in line 173.
Responses: Thank you for this suggestion. And we have corrected this mistake now in line 199. “where N represents the total number of years in the study, and Ai and Bi represent the ET products used in this study and the water balance ET (ETw) respectively.”
20) Define ETw prior to the line 174.
Responses: Thanks. We have followed your suggestion and add a definition in the line 181 now, “where ETw is the ET calculated by the water balance method”.
21) The symbols of ETw in equations and text are not consistent. While in equation it has ETw but in text ETw.
Responses: We are grateful for this suggestion. We have discovered this problem and unified the format of ETw as ETw in the full text. Thank you very much.
22) Figure 4 needs better presentation. If possible, please make such graph either with secondary axis or make separately. It is not clear the purpose of this figure as well.
Responses: Thank you for this suggestion. In order to make the purpose of this picture more obvious, we introduced GRACE data in the period of 2004-2012.The GRACE data was processed in CSR and we have performed stripe filtering and Gaussian filtering in order to reduce the uncertainties in estimating ΔS. Then, We compare the total soil moisture change (TSMC) with TWSC (ΔS) estimated by GRACE in the period of 2004-2012 (Fig 4,Right) to confirm that the consistency between the two is very well and ΔS is still very small after 2002. So, through Figure 4, we can obtain that neglecting ΔS is credible in the water balance method in Yellow River Basin. Besides, we also have changed figure 4 into double y-axis line plot.
Figure 4 (Left) The time series of annual soil moisture changes (TSMC) and the differences in precipitation and runoff (P-R) for the Yellow River Basin during 1982-2000 and (Right) the comparision between terrestrial water storage changes (ΔS) estimated by GRACE and annual soil moisture changes (TSMC) during 2004-2012.
23) The values of BIAS and RMSD for E2O product reported in line 283 do not matched with ones included in Table 4. It is not sure which one is the right. Please revise them. Similar mistakes are repeated for GLDAS in lines 287 and 288
Responses: Appreciate you for the suggestions. We are sorry that this is an error that should not occur. We have corrected all the wrong parts and now all the values are matched.
24) A portion of the sentence started in line 289 "the BIAS are -56.12... respectively seems to be repeated.
Responses: We are grateful for this suggestion. We are sorry that this is a very careless mistake, and we have removed the duplicates.
25) In line 294, article "a" shall be replaced with "the"
Responses: Thank you very much for this suggestion and sorry for that this is a very careless mistake. And we have corrected it with you suggestion in line 310, which reads “appears have the highest correlation coefficient of 0.86 with the ETw”.
26) In Figure 5, authors presented the interannual variations of land ET from different ET product, but it is not mention that which catchment these results belong to.
Responses: Thanks. The catchment we selected is the catchment whose outlet is Huayuankou station. We have pointed out it in the revised manuscript in line 293-295, which reads “To understand the temporal variations of land ET, we show in Figure 5 the annual time series of the five ET products and the ETw estimated from the water balance method across the Yellow River Basin controlled by Huangyuankou station.” The reason why we choose the catchment controlled by Huayuankou instead of that by Lijin is because the throttling and shut-off phenomena in the lower Yellow River are very serious (Huang et al. 2003), so the uncertainty in the catchment controlled by Lijin is even greater.

Reviewer 2 Report
Manuscript water – 395135 “Evaluation of evapotranspiration estimates in the Yellow River Basin against water balance method”
All over the article, the experimental data period considered is 1982 to 2000, an 18 years period. It is too short, if compared to the 30 years period usually considered for climatological characterization and other studies, as recommended by the UN World Meteorological Organization. As climate change imposes with increasing evidence, more reasons come up to use data referring to larger periods. In the present case, it is not clearly stated weather a shorter period was chosen or imposed by available data, including base data on precipitation and downward short-wave radiation, also used as forced data for complementary analysis. If available, a longer period would better be considered. If not, doubt remains about the reasons for the observed performance levels of some models/products studied.
The term “product” is applied to each ET calculation system in comparison within the study. Why? More usually, the term “model” is used in such cases …
The 5 “products”/models studied should be formerly described and presented, in brief (v.g. after line 79).
It could also be briefly justified why use these calculation “products”, not simply Penman-Monteith or derived models, as is more usual, under FAO recommendations.
I would also suggest the authors to review the Discussion and Conclusions sections, mainly in what justifies further study of each “product” with specific set of forced data.
A careful edition of the English text is also necessary.
Author Response
We are grateful to all the reviewers for their helpful comments to improve the manuscript. All the comments have been carefully read, and we have responded to them as suggested by the reviewers.
Comments from reviewer 2:
1) All over the article, the experimental data period considered is 1982 to 2000, an 18 years period. It is too short, if compared to the 30 years period usually considered for climatological characterization and other studies, as recommended by the UN World Meteorological Organization. As climate change imposes with increasing evidence, more reasons come up to use data referring to larger periods. In the present case, it is not clearly stated weather a shorter period was chosen or imposed by available data, including base data on precipitation and downward short-wave radiation, also used as forced data for complementary analysis. If available, a longer period would better be considered. If not, doubt remains about the reasons for the observed performance levels of some models/products studied.
Responses: We acknowledge and agree with the point raised by the reviewer in terms of long term data for better characterization of the climatology. The study period (1982-2000) seemed too short compared to the 30 years period usually considered for climatological characterization. Unfortunately, the runoff data in the Yellow River Basin is only available for time period of 1956-2000. However, according to the time range of each product (1982-2012), we decided to set the study period to 1982-2000. This limitation is added to the revised manuscript in line 227-228, which reads as,
“However, it is worthwhile noticing that the results are not robustly conclusive and have great uncertainty, due to the time-span of the data.”
2) The term“product”is applied to each ET calculation system in comparison within the study. Why? More usually, the term“model”is used in such cases …
Responses: We appreciate the reviewer inputs in order to improve the quality of the work. Before we conducted this study, we have reviewed numerous papers, in which the term“product”was used in such cases. Additionally, these refer to outputs of models and not insitu measurements. Furthermore, we believe that the term“product”and“model”are not of the same semantic. The products are the outputs of the models, and this is what we use for the analysis.
3) The 5“products”/models studied should be formerly described and presented, in brief (v.g. after line 79).
Responses: We acknowledge the reviewer’s suggestion about the detailed description of the products/models. In the revised manuscript we have described 5 ET products in more details which can be read from line 124-150.
4) It could also be briefly justified why use these calculation“products”, not simply Penman-Monteith or derived models, as is more usual, under FAO recommendations.
Responses: Thank you for this comment. The biggest advantage of these products is that they can be used to study the water cycle problem at the regional or basin scale, and the time range of these is long enough. ET estimates from point observations, such as eddy covariance systems, Bowen ratio energy balance systems and lysimeters, are based on short durations and have limited spatial coverage (Mao et al.,2017). This, we also point out it in the reviewed manuscript in line 33-35.
“Currently, ET is mainly measured from site observations, using lysimeters [7], the Bowen ratio energy balance, and flux towers [1]. But such measurements are based on short durations and have limited spatial coverage”
5) I would also suggest the authors to review the Discussion and Conclusions sections, mainly in what justifies further study of each “product” with specific set of forced data.
Responses: The suggestions made by the reviewer is indeed very precious and valuable for the suggesting future studies in a specific direction. In this work, our main purpose is to study the performance of various products in the Yellow River Basin. Here, we simply analyze the impact of the main forcing data (precipitation and radiation) of each product and we admit that these may be not enough for understanding the relationship between the products and their specific forcing data. Diagnosing such issues is beyond the scope of the current work which however is highlighted in the conclusion section for future studies in line 451-453, which read as,
“And in the future work, we should pay more attention on the uncertainties in the ET models and improving the quality of forcing datasets, as the better forcing dataset is necessary to produce more credible ET products with better performance.”
6) A careful edition of the English text is also necessary.
Responses: We acknowledge the reviewers time and inputs for these suggestions and improvements. The manuscript is edited for grammar and technical mistakes from professional English language editing services. The certificate is attached with the revised manuscript for consideration. Further, we have carefully reviewed this article and have removed many careless errors and formatting errors.

Reviewer 3 Report
The review is attached as .pdf document

Author Response
We are grateful to all the reviewers for their helpful comments to improve the manuscript. All the comments have been carefully read, and we have responded to them as suggested by the reviewers.
Comments from reviewer 3:
1) The paper was interesting to read, it is well written and structured, but the paper does not introduce significantly new results. In the paper the authors analyze the data from 1982-2000, avoiding to include the fresh one.
Responses: Thank this reviewer very much for the positive comments and the constructive suggestions, and the point-by-point responses are shown below. In order to ensure the accuracy of the data, we chose the most accurate runoff data in the Yellow River Basin which is obtained from the Yellow River Conservancy Commission. But its time range is 1956-2000. According to the time range of each product (1982-2012), we finally set the study period to 1982-2000
2) The most important weakness. The authors assume ΔS, terrestrial water storage, to be zero averaging over time. The assume is not well supported. To see temporal evolution (and interannual variations) of ΔS, I suggest to make new figure 4 including period after 2002, with ΔS calculated using GRACE data, regardless overlapping of periods (lines 158-160). Also, Figure 4 needs improvements. I suggest removing bars, and making double y-axis line plot. Please add the additional text defending the choice for ΔS, with discussion of spatial/temporal changes with reflection on line255-257. If ΔS is not zero the analysis has minor importance.
Responses: We thank this reviewer very much for these suggestion, which can greatly improve this study to be more complete and rigorous. We have followed your suggestion and used the GRACE data to better represent the temporal evolution of ΔS. The GRACE data was processed in CSR and we have performed stripe filtering and Gaussian filtering in order to reduce the uncertainties in estimating ΔS. Then, We compare the total soil moisture change (TSMC) with TWSC (ΔS) estimated by GRACE in the period of 2004-2012 (Fig 4,Right). And results show that the consistency between the two is very well and ΔS is still very small compared to ET in the period after 2002. Thus, neglecting ΔS is credible. Furthermore, we have changed figure 4 into double y-axis line plot.
Figure 4 (Left) The time series of annual soil moisture changes (TSMC) and the differences in precipitation and runoff (P-R) for the Yellow River Basin during 1982-2000 and (Right) the comparision between terrestrial water storage changes (ΔS) estimated by GRACE and annual soil moisture changes (TSMC) during 2004-2012.
3) Statistical analysis, correlation coefficient, is rudimental tool (equations 2-5 are well known, and very similar metrics. I suggest to remove it from the manuscript. Also equation 6.).
Responses: Appreciate you for the suggestion. We have followed your suggestion to remove the equations 2 and 6, but we have reserved equations 3-5. Because we think that although they are well known, there is still a need to show them in order to make the structure of the article complete, as many articles did before (Mao et al.,2017; Liu et al. 2016; Xue et al.2013; Li et al.,2017).
4) The final time series data included in analysis have 12 points in spatial and 18 points in temporal domain, making conclusions very weak. Correlation in fig2 is statistically significant but I thing strongly depends (ZHANG, E2O, GLDAS) on two extreme data ETw=350 mm/y and 530 mm/y. Please comment!
Responses: Appreciate you for the suggestions, which can improve our study to be more rigorous. We have realized that the conclusions are very weak due to very limited study data. This is the uncertainty in statistics and we have already pointed out this uncertainty in the reviewed manuscript in line 227-228, which reads “However, it is worthwhile noticing that the results are not robustly conclusive and have great uncertainty, due to the limited amount of data.” Besides, the two extreme values do have some impact, but after we tried to delete the two values, the results did not change much.
5) In the light of my comments, the paragraphs (line 223-246; 275-309; 379-420) are too long. I suggest making them shorter, more compact and relevant/significant. In table 4 correlation for ZHANG (0.33) is not statistically significant (p=0.05). Line 293-297, please comment in the light of short time series and basic statistical analysis.
Responses: We are grateful for this comment. We have improved these paragraphs according to the reviewer’s comments, and the detailed revision can be found the revised manuscript. And in the analysis of interannual variations, we pay more attention to commenting in the light of short time series and make significant tests on correlation coefficients. And results show that only the correlation for ZHANG is not significant. In the reviewed manuscript, it reads “The ZHANG product, which performs the best in terms of the magnitude, appears to perform the worst in terms of interannual variations in the period of 1982-2000 and it is the only product that does not pass the significance test at 0.05 level.”
6) Improve figure 1 – change miles in kilometers, remove 00’, show and expand surrounding areas with meteorological stations and shortwave radiation stations (line 379-383). It is necessary to understand spatial errors in variables (line 422).
Responses: Appreciate you for the suggestion. And we have followed your suggestion to improve figure 1 with adding the location of meteorological stations and solar radiation sites in it. And we also change miles in kilometers, remove 00’.
7) Please use scientific number format avoiding “,” as digital grouping symbol.
Responses: Thank you for your comments. But we use “,” as digital grouping symbol because this is the format requirement in “Water” and we have seen similar usage in the articles published in Water before (Deng et al., 2018; Fry et al., 2018).
1.Deng, X.H.; Song, X.Y.; Xu, Z.M. Transaction Costs, Modes, and Scales from Agricultural to Industrial Water Rights Trading in an Inland River Basin, Northwest China. Water. 2018.
2.Fry, T.J.; Maxwell, R.M. Using a Distributed Hydrologic Model to Improve the Green Infrastructure Parameterization Used in a Lumped Model. Water. 2018, 10,1756.
8) Section 2.2.2 – How many meteo stations are in the investigation area (Figure 1)?
Responses: According to this comment, we have showed the location of meteorological stations and solar radiation sites in figure 1. And there are 307 meteorological stations and 79 solar radiation sites in the investigation area. Notably, the scope of the original precipitation and radiation data is the whole of China, which integrated from 2472 meteorological stations and 719 solar radiation sites, we intercepted the Yellow River Basin data for comparison.
9) What is temporal coverage of precipitation and streamflows, gaps (missing data) in the time series? I suggest making the additional figure showing monthly means time series of all streamflows and characteristic precipitation measurements and combine it with figure 6.
Responses: Thank you for your comments. We have followed your suggestion and added a new figure about monthly means time series of streamflows and precipitation in line 434. And results show that the runoff data is consistent with the precipitation and have obvious seasonal characteristics, indicating that the precipitation has a significant role in the water cycle in the Yellow River Basin.
Figure 7 The monthly precipitation and runoff data across the Yellow River Basin
10)Figure 5 is difficult to analyze, please follow the suggestion for figure 1. It will be interesting to put trend on the figure.
Responses: According to this comment, we have changed figure 5 into double y-axis line plot and put trend on the figure.
11)Please consider to remove Table 5 and supporting text in the 2.2 section.
Responses: Appreciate you for the suggestion. And we have removed Table 5 and supporting text in the 2.2 section.
12)I suggest adding “Conclusion” section with 4.2 paragraph in it.
Responses: Thank you for your suggestion. According to this comment, we have added conclusions and proposed some future work directions in the revised manuscript in line 442-450. We also include 4.2 paragraph in the conclusion.
13)I suggest removing the tables 3 and 4 from the manuscript because there is no important information’s.
Responses: We are grateful for this comment. We think the results of two indicators (BIAS and RB) are too similar and table 3 has no important information, so we follow your suggestion and remove the result of BIAS and table 3. But we think the original table 4 (table 3 now line 301-303) is a very important table and it is the main supplement of figure 5, so we decide to reserve it.
14)Please avoid the term “clearly seen”, line 265.
Responses: Thank you for your suggestion. We have followed your advice and removed the term “clear seen” as well in line 270.
15)Line 144-146, please expand the sentence.
Responses: Thank you for your comments. We have expanded the sentence to introduce the soil moisture data we used more specifically in line 150-154, which reads, “The soil moisture data is obtained from the Land Surface Processes and Global Change Research Group (http://hydro.igsnrr.ac.cn/public/vic_outputs.html). And the data is derived from the simulations of Variable Infiltration Capacity (VIC) model forced by meteorological observations with a 3-hour time step and a 0.25° spatial resolution during the period of 1952-2012, which have been demonstrated as a more reliable soil moisture data in the China ”.
16)Line 293-297: Please comment
Responses: Thank you for your comments. And we have payed more attention to commenting in the light of short time series and make significant tests on correlation coefficients. In the reviewed manuscript in line 300-305, it reads “The ZHANG product, which performs the best in terms of the magnitude, appears to perform the worst in terms of interannual variations in the period of 1982-2000 and it is the only product that does not pass the significance test at 0.05 level. In contrast, the E2O product, which shows the worst performance in terms of magnitude, appears have the highest correlation coefficient of 0.86 with the ETw, indicating that it best describes the interannual variations among the five ET products.”

Round 2
Reviewer 1 Report
The reviewer sees that the authors have greatly improved the manuscript by modifying figures and tables together with substantial amount of new texts. The current version is more likely to get published once the authors’ responses the minor revisions.
While most of the comments are carefully addressed, the authors, however, poorly addressed the suggestions: (Points 1, 7 and 15-17). These section some modifications as follows.
It is good to take care of issues on the structure of the paper by adding Conclusion at the end. However, it is not still clear what are the discrepancies in the performance of five ET products.
The statement made in lines 43-48 is not so much clear as it is not mentioned what are those features make more sense to understand the processes. What are the physical phenomenon or reasoning behind this? Please be implicit.
The response to #15 still requires the method to interpolate annual precipitation across the basin implicitly.
It is good to have brief explanation of MTE. But rather introducing new jargon may make reader confuse. Please be careful.
The comment #17 has been wisely taken care of but in such issues, it will be great if citation is added in the last.
To the end, the reviewer congratulate to the authors and hoping to see much improvement in the final version.
Author Response
Comments from reviewer 1:
1) It is good to take care of issues on the structure of the paper by adding Conclusion at the end. However, it is not still clear what are the discrepancies in the performance of five ET products.
Response: We appreciate the reviewer’s comments. We have written new conclusions and added the discrepancies in the performance of five ET products in the conclusion. This can be seen in the revised manuscript on page 16 lines 444-463 as,
“The significance of the Yellow River Basin to both the settlements over the region and China as a whole, as well as to the climate has been shown in previous studies and this study. ET, as a climate process, has a distinct impact on the hydrologic and energy cycle of the basin, consequently impacting the available water over the region. The difficulty in measuring ET as well as the lack of available observations has made the use of additional sources of ET information from reanalysis and satellite products a very important option. However, different sensitivities of these additional existing products to climate variables such as precipitation and net radiation has created the necessity to evaluate them and understand their strengths and differences. In this study, three commonly used independent ET datasets over the basin, the ZHANG, GLDAS and MERRA_Land are evaluated against water balance method to understand how they capture ET processes. Additionally, the GLEAM and E2O, which have never been evaluated in the region were also used in this study. While all the datasets were found to have captured the temporal dynamics of ET over the basin, the GLEAM product showed significantly close temporal variabilities to the ET estimated with water balance method. The MERRA_land product performed best in describing the spatial characteristics and the GLDAS product had a satisfactory performance as well. The ZHANG product was also found to show very close magnitudes to the reference ET while the E2O product best described the interannual variations. Due to the limited data available, there still remains uncertainties in the representation of the long-term climatology of ET in the basin. With the continuous production of additional data, future studies will be able to provide more credible information to this field.”
2) The statement made in lines 43-48 is not so much clear as it is not mentioned what are those features make more sense to understand the processes. What are the physical phenomenon or reasoning behind this? Please be implicit
Response: We acknowledge the reviewer’s inputs for indicating this point which indeed is confusing and misleading. And in the revised manuscript, we have added some specific examples to make it clearer, which can be seen on page 2 line 43-47 as,
“This has enabled us to better understand processes in such regions at large scale rather than point scale, for example, MERRA Improved our understanding of the water cycle by considering various aspects of the hydrological cycle [13] and JRA is suitable for multidecadal variability and climate change as it is the first reanalysis data whose study time covering the last half-century [14].”
3) The response to #15 still requires the method to interpolate annual precipitation across the basin implicitly.
Responses: We acknowledge the reviewers comment for improving the quality of the work. In the modified manuscript we have introduced the interpolation method on page 5 line 156-163 which read as,
“For Precipitation, we used 19 years of observed precipitation data (Gridded Precipitation 0.5°×0.5° Grid Dataset, V 2.0) developed by China meteorological Administration (CMA) and National Meteorological Information Center (NMIC) (http://cdcNaNa.gov.cn/). The dataset is developed from daily precipitation records of 2474 meteorological stations across the country using Thin-plate spline [37] and GTOPO30 (Global 30 arc-second elevation) DEM data for reducing the influence of elevation. Different studies have used the data for climatological applications with good degree of agreements [38]. Further details about gridded precipitation data are available in NMIC (2012).”
4) It is good to have brief explanation of MTE. But rather introducing new jargon may make reader confuse. Please be careful.
Responses: Thank you very much for this suggestion, these expressions indeed can be confusing for the readers. The new jargon “match-learning” is our mistake and we have corrected it into “machine-learning”. Page 2 line 47-51,
“Among them, the MTE (Model Tree Ensemble) product, based on eddy covariance, is derived from in-situ observations by a machine-learning algorithm and it is considered as high quality accurate ET product and have been widely used as a standard to validate other products. But it is limited to a relatively short period and has a sparse spatial coverage [8,15].”
5) The comment #17 has been wisely taken care of but in such issues, it will be great if citation is added in the last.
Responses: Thank you very much for this comment. And we have put these citation in the last. Page 2 line 70-77,
“For example, Liu conducted a global comparison of nine ET products against the water balance method during the period of 1983-2006 [8]; it is found that all products could not explain the long-term trends accurately, especially in the wet basins. Zhang reached a similar conclusion in 110 wet basins at global scale [9]. Li and collaborators evaluated the seasonal changes of various ET products on the Tibetan Plateau, using the water balance method [23]; and found that the analyzed ET products had a great difference in performance among them. Li compared nine ET products in the middle Yellow River Basin and found that the ET derived from LSMs performed better than the reanalysis data [26]. ”

Reviewer 3 Report
Dear
Thank you for answering on my comments.
For me, the manuscript is significantly improved.
My recommendation is to publish the manuscript in the present form.
Best regards
Author Response
Thank you very much for your positive comments.